

# Integration of the Global Water and Lake Sectors within the ISIMIP framework through scaling of streamflow inputs to lakes

Ana I. Ayala[1], José L. Hinostroza[2], Daniel Mercado-Bettín[3], Rafael Marcé[3], Simon N. Gosling[4], Donald C. Pierson[1], Sebastian Sobek[1]

[1]Limnology Unit, Department of Ecology and Genetics, Uppsala University, Uppsala, Sweden.
[2]Faculty of Civil Engineering, National University of Engineering, Lima, Perú.
[3]Centre for Advanced Studies, National Spanish Research Council (CEAB-CSIC), Blanes, Spain.
[4]School of Geography, University of Nottingham, Nottingham, United Kingdom.

*Correspondence to*: Ana I. Ayala (isabel.ayala.zamora@ebc.uu.se)

**Abstract.** Climate change impacts both lakes and their surrounding catchments, leading to altered discharge and nutrient loading patterns from catchments to lakes, as well as modified thermal stratification and mixing dynamics within lakes. These alterations affect biogeochemical processes and water quality in lakes. Coupled catchment-lake modeling provides both a holistic evaluation of the effects of climate change on lakes and a framework for explicitly assessing the importance of how catchments effect lakes. The Inter-Sectoral Impact Model Intercomparison Project (ISIMIP) provides a framework for projecting the impacts of climate change across multiple sectors (e.g. water, lakes, energy, health) of the Earth System consistently, enabling integrated cross-sectoral assessments. However, climate impacts on lake dynamics are modeled in ISIMIP without consideration of the links between lakes and the surrounding catchments. This is a significant limitation, as it restricts assessments to only the direct impacts of climate change on lakes, overlooking the critical interactions between lakes and their catchment areas. In this study, we establish the first dynamic connection between the Global Water and Lake Sectors in ISIMIP, achieved by scaling the gridded modeled outputs of water fluxes from the Global Water Sector to the catchments of the representative lakes of the Lake Sector. The streamflow to the representative lake of each grid cell, as defined by the ISIMIP Global Lake Sector, was calculated based on runoff proportional to the catchment area of each representative lake. If the lake surface area was larger than the grid cell area, water from upstream grid cells was included as the corresponding proportion of river discharge. The methodology was applied to 71 lakes of widely different size across Sweden, and the estimated streamflow was validated against both the streamflow outputs from the hydrological model HYPE and observed data. Our procedure showed good performance in terms of long-term streamflow mean and seasonality, with a yearly average Kling-Gupta efficiency, KGE, of 0.54±0.23 and a monthly average KGE of 0.59±0.18 when compared to HYPE outputs, and with yearly and monthly average KGEs of 0.73±0.16 and 0.50±0.19, respectively, when compared to observations. This estimated streamflow, representing water flow into lakes, will provide a valuable dataset for the scientific community within the ISIMIP Lake Sector supporting hydrological and water quality modeling efforts aimed at understanding the impacts of climate change on lakes.



## 1 Introduction

Climate change impacts both catchments and lakes in distinct yet interconnected ways, influencing their physical, chemical, and biological processes. On one hand, alteration of precipitation patterns and increases in air temperature lead to hydrological

changes in catchments. At high latitudes, winter precipitation increases and increased air temperature shift precipitation from snow to rain, reducing snowpack and leading to earlier spring snowmelt, resulting in greater streamflow and nutrients loading during this period (Jiménez-Navarro et al., 2021). Higher air temperatures also lead to greater evapotranspiration rates (Donnelly et al., 2017; Liu et al., 2021), which reduce streamflow and increase nutrient concentration during the summer. Extreme precipitation leads to increased nutrient loading in both wet and dry areas, through increased runoff in wet areas, and

soil erosion and the mobilization of nutrients trapped in soils in dry areas (Costa et al., 2023). On the other hand, increases in air temperature result in increased lake surface water temperature (O'Reilly et al., 2015) with stronger thermal stratification and reduced mixing (Kraemer et al., 2015), earlier onset of summer stratification (Magee and Wu, 2017; Moras et al., 2019) and shorter ice-cover periods (Sharma et al., 2019, 2021). Warmer water temperatures promote the growth of cyanobacteria, leading to the formation of harmful algal blooms (Paerl and Huisman, 2008; Huisman et al., 2018).

Stronger lake stability and longer duration of thermal stratification lead to hypolimnetic oxygen depletion (Jane et al., 2021; Jansen et al., 2024), resulting in increased internal loading (North et al., 2014) and greenhouse gas emission (Marotta et al., 2014; Vachon et al., 2019; Jansen et al., 2022). Earlier ice loss leads to greater heat loss due to increased evaporation rates (Wang et al., 2018; Li et al., 2022). Nonetheless, climate change affects lake ecosystems through a complex and dynamic interplay of catchment loading and lake-internal processes. For instance, changes in the timing of streamflow and lake ice-off

lead to earlier onset of spring phytoplankton blooms (Gronchi et al., 2021; Mesman et al., 2024). Lake water level fluctuations caused by dry conditions during periods of strong stratification limit vertical mixing, which contributes to pronounced hypolimnetic hypoxia. This hypoxia is exacerbated by intense precipitation events that subsequently reduce oxygen concentrations in the water column and lower the pH level (Saber et al., 2020).

The integration of catchments and lakes using coupled dynamic models provides valuable insights into the functioning of lake

ecosystems and the impacts of climate change, for example to understand how changes in the surrounding catchment area, within the lake itself, and their interactions affect the lake's dynamics. Additionally, this modeling approach can inform the development of adaptation and mitigation strategies.

The Inter-Sectoral Impact Model Intercomparison Project (ISIMIP, https://www.isimip.org) is a collaborative framework for assessing the impacts of climate change across temporal and spatial scales, by integrating climate models, impact models, and

direct human forcing data to provide insights into climate change risk and inform potential adaptation and mitigation strategies. ISIMIP is organized into multiple sectors that represent natural and human components of the Earth system, that are both regulators of climate and vulnerable to its changes, including agriculture, forests, fisheries and marine ecosystems, water, lakes, energy, health, among others. To ensure consistency in impact modeling within and across sectors, ISIMIP provides a common set of climate-related and direct human forcing data, and along with a modeling protocol sets up standardized




experiments, spanning pre-industrial and historical periods and future projections (Frieler et al., 2024). This framework enables multi-model impact simulations within sectors, enhancing the robustness and reliability of model projections (Rosenzweig et al., 2017) and quantifying the sources of uncertainty in the projections (Krysanova et al., 2017; La Fuente et al., 2024a; Jones et al., 2025). In addition, it enables cross-sectoral assessment of climate change impacts (Lange et al., 2020; Vanderkelen et al., 2020). However, cross-sectoral integration remains a challenge within ISIMIP, which limits the potential for capturing

complex interdependencies, cascading effects, and feedback loops between sectors. For example, simulations from the Water Sector and the Lake Sector are at present not connected to each other.

   The ISIMIP Lake Sector modeling has focused on lake physics and thermal dynamics, including changes in water temperature (Ayala et al., 2020, 2023b), loss of ice cover (Grant et al., 2021; Sharma et al., 2021), stratification phenology (Woolway et al., 2021b, 2022b), alterations in mixing regimes (Woolway and Merchant, 2019), occurrence of lake heatwaves (Woolway et

al., 2021a, 2022a), shifts in lake thermal regions (Maberly et al., 2020), heat uptake (Vanderkelen et al., 2020), surface heat fluxes (Ayala et al., 2023a) and lake evaporation (La Fuente et al., 2022, 2024b, a). Hydrodynamic lake model simulations were performed under the premise that lake water temperature variation results solely from the exchange of energy between the lake surface and the atmosphere (Golub et al., 2022). However, the advective fluxes are particularly relevant for water bodies with significant water level fluctuations and rapid water exchange, such as reservoirs or lakes with short residence

times. Fenocchi et al. (2017) showed that in a deep subalpine lake a hydrodynamic lake model, when excluding through-flows, required unrealistically low light extinction coefficient to reproduce temperatures in the epilimnion and upper metalimnion. In contrast, incorporating through-flows together with a realistic light extinction coefficient improved the accuracy of temperature predictions in the lower metalimnion and upper hypolimnion. Råman Vinnå et al. (2018) investigated the tributary influences on lakes in a study of Lake Biel and Lake Geneva and revealed that seasonal variations in river discharge and temperature

significantly affect lake warming and stratification, underscoring the importance of hydrologic inputs in thermal lake modeling. Integrating the ISIMIP Water Sector and Lake Sector, incorporating hydrologic model outputs into lake model simulations, can improve the accuracy of thermal stratification and mixing dynamics in lakes and the assessment of climate change impacts. It will also provide the basis for more complex simulations of lake biogeochemistry and water quality parameters, for which inputs from the upstream catchment are paramount.

The Global Water Sector in ISIMIP (Telteu et al., 2021; Müller Schmied et al., 2024a) focuses on assessing the impacts of climate change on water fluxes, including discharge, total (surface + subsurface) runoff and evapotranspiration, among other hydrological variables, with a global grid resolution of 0.5° by 0.5°. The Global Lake Sector in ISIMIP has assigned one representative lake to each 0.5° grid cell, which simplifies the complexity of modeling all lakes globally. This ensures computational feasibility while capturing variations in lake responses to climate change, and provides a practical way to include

lake-specific dynamics in a global-scale assessment.

   Here, gridded water fluxes simulated by WaterGAP 2 following the ISIMIP phase 3a protocol were scaled to match the individual lake catchment areas for estimating the streamflow of 71 lake catchments across Sweden. The catchment-scale streamflow simulations were then validated against both the streamflow outputs of the hydrological model HYPE, which



simulates hydrological processes at the catchment scale, and observed data. The use of HYPE provides an additional
benchmark, with its outputs serving as an established reference dataset where observational data are limited.

## 2. Material and methods

### 2.1. Gridded simulations of streamflow from the ISIMIP3a Global Water Sector

Water Global Assessment and Prognosis (WaterGAP) is a process-based hydrological model used for quantifying water resources and water use on a global scale (Alcamo et al., 2003; Döll et al., 2003). WaterGAP 2 consists of three major components, the global water use model, the linking model GroundWater-SurfaceWater USE (GWSWUSE) and the WaterGAP Global Hydrology Model (WGHM) (Müller Schmied et al., 2021). The global water use model distinguishes five water use sectors, i.e., irrigation, livestock, domestic, manufacturing and cooling of thermal power plants, quantifying both consumptive waters use and water withdrawals. The linking model GWSWUSE computes the fractions of water withdrawals and consumptive use for all five sectors, distinguishing whether the water is sourced from groundwater or surface water bodies, such as lakes, reservoirs and rivers. The WGHM computes water flows (fast surface and sub-surface runoff, groundwater recharge, evapotranspiration and river discharge) and storage across ten compartments. The vertical water balance covers the canopy, snow and soil, while the lateral water balance includes groundwater, lakes, reservoirs, wetlands and rivers.

The computational grid of WaterGAP 2 is based on the CRU land-sea mask (Mitchell & Jones, 2005), which covers the global continental area with the exception of Antarctica, comprising 67420 grid cells of 0.5° longitude x 0.5° latitude, and the upstream–downstream relations among the grid cells are defined by the drainage direction map DDM30 (Döll and Lehner, 2002). Model input includes climate data, land use and land cover data, soil characteristics, location and extent of surface water bodies (lakes, wetlands, dams and reservoirs), the river routine network (basins, flow direction and slopes) and human water use data. Although WaterGAP 2 includes the representation of lakes in its simulations (Müller Schmied et al., 2021), accounting for their role in storing water, evaporation and downstream release, it does not explicitly resolve the amount of water flowing into individual lakes from upstream locations. Instead, the model estimates water flows at the grid cell level, without disaggregating them to accurately capture inflow to specific lakes. As a result, it lacks the spatial resolution needed to track lake specific inflow in details.

WaterGAP 2.2e contributed to ISIMIP3a, specifically in the Global Water Sector (Müller Schmied et al., 2023), following the ISIMIP3a simulation protocol. The protocol (https://protocol.isimip.org) outlines the required experiments, input data sets and output variables necessary for participation. Input data (climate forcing, socioeconomic forcing and static geographic information) and impact model outputs are available at https://data.isimip.org.

Here, we focused on the standard model evaluation experiment *obsclim_histsoc_default* (Frieler et al., 2024), which is based on the observed climate-related forcing *obsclim* from the GSWP3-W5E5 climate forcing dataset (Kim, 2017; Cucchi et al., 2020; Lange et al., 2021) combined with the direct human forcing (e.g. land use and land cover changes and water



management) *histsoc*. This experiment reproduces observed long-term changes in hydrological change and water use from 1901 to 2019. The impact model WaterGAP 2.2e (https://www.isimip.org/impactmodels) provided monthly total (surface + subsurface) runoff, $q_{tot}$ [kg m$^{-2}$ s$^{-1}$], groundwater runoff, $q_g$ [kg m$^{-2}$ s$^{-1}$], and discharge, *dis* [m$^{-3}$ s$^{-1}$]. Note that, $q_{tot}$ and $q_g$ represent the runoff produced within a grid cell, which includes both surface and subsurface, and groundwater components. Meanwhile, *dis* represent the routed (i.e. via river channels) discharge at a given grid cell, which includes the runoff generated

within that grid cell plus the contribution from upstream grid cells. This means *dis* accounts for both locally generated runoff and the cumulative flow from upstream grid cells, which results in the total discharge flowing downstream.

## 2.2 Representative lakes in the ISIMIP3a Global Lake Sector

In the ISIMIP3a Global Lake Sector, each 0.5°grid cell was assigned a representative lake sourced from the HydroLAKES database (Messager et al., 2016). The selection of the representative lake for each grid cell was based on the location of the

lake centroid. When multiple lakes were present within a grid cell, the representative lake was selected based on the lake depth corresponding to the weighted median of all lakes within the respective grid cell (weighted by the area of the lake) (Golub et al., 2022), resulting in a total of 41449 representative lakes. The catchment areas for these representative lakes were derived from the HydroLAKES database (Messager et al., 2016).

## 2.3 Study sites

Our study sites comprised 71 lakes across Sweden, which correspond to representative lakes from the ISIMIP3a Global Lake Sector (Figure 1). The site selection included lakes with a wide range of surface areas and catchments of varying sizes (Table S1). Additionally, the lakes were located in different physiographic regions, ranging from agriculturally dominated lowland areas in the south to boreal and subarctic regions in the north. Accordingly, the surface area, $A_{lake}$, ranged from 0.34 to 5486 km$^2$, with mean and median values of 173.89 and 23.14 km$^2$, respectively. The catchment area, $A_{catchment}$, varied from 101 to

48421 km$^2$, with mean and median values of 4481 and 1662 km$^2$, respectively. The ratio between $A_{catchment}$ and $A_{lake}$ ranged from 3.37 to 14962. The mean and median ratio $A_{catchment}$ to $A_{lake}$ were 381.89 and 45.96.



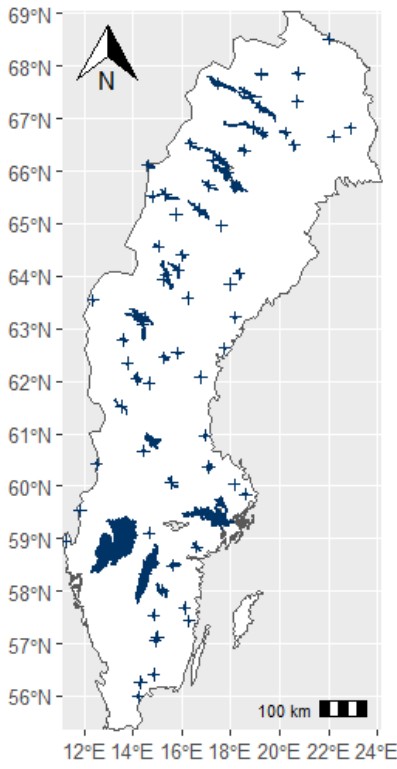

**Figure 1. Study sites, marked either as crosses (small lakes) or blue lake shapes (large lakes).**

## 2.4 Scaling streamflow from 0.5°grid cells to catchment scale

To estimate the streamflow into representative lakes, we applied a scaling approach that adjusts grid-based hydrological outputs to the actual catchment area of the lakes. The method is structured into three approaches, depending on the size of the catchment relative to the grid cell area where the lake centroid is located:

- Approach I.a: Applied when the catchment area is smaller than or equal to the area of the grid cell containing the lake centroid.

- Approach I.b: Applied when the catchment area spans multiple grid cells.
- Approach II: Applied for large lakes where the lake area spans multiple grid cells.

The water flow into the representative lake was calculated based on total (surface + subsurface) runoff, $q_{tot}$ [m s$^{-1}$], and groundwater runoff, $q_g$ [m s$^{-1}$], in proportion to the catchment area of the representative lake, $A_{catchment}$ [m$^2$] (Figure 2). The catchment was delineated using upstream grid cells based on the flow direction. The grid cells contributing water flow towards

the lake were classified into levels: grid cells partially occupied by the lake correspond to level 0; each level 0 grid cell received water flow from one or more of its eight neighbouring grid cells, which were classified as level 1; this process continued, with the neighbouring grid cells of level 1 classified as level 2, etc (Figure 4). The ratio ($N$) of catchment area, $A_{catchment}$, to grid cell





area (grid cell area where the lake centroid is located), $A_{grid}$, indicated how many grids cell occupied the catchment and determines the number of grid cells to be counted in the water flow calculation (Figure 2).

Figure 2. Workflow for estimating streamflow to lakes by scaling grid cells to catchment scale.

For $N \leq 1$ (Approach I.a), the catchment area was smaller or equal to the grid cell area where the lake centroid was located. Note that, $N$ was rounded down to the nearest integer number, meaning the catchment area can be slightly greater than the lake grid cell area. Only grid cells partially occupied by the lake (grid cells of level 0) were counted ($i$) (Figures 2 and 3). For example, lake 12247 (Figure 3) had an $A_{lake}$ of 22.46 km$^2$ and partially occupied 2 grid cells. $A_{catchment}$ was 1662 km$^2$, which was slighter greater than the $A_{grid}$ of the grid cell where the lake centroid was located, which was 1415.32 km$^2$, resulting in a $N$ equal to 1. Therefore, the grid cells occupied by the lake (grid cells of level 0) were included in the count ($i = 2$).





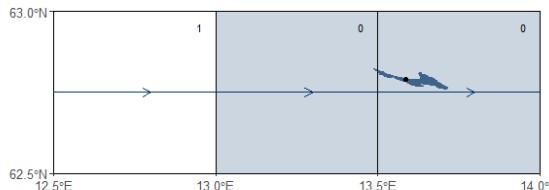

**Figure 3. Catchment-lake scheme for lake 12247 ($A_{lake}$ = 22.46 km², $A_{catchment}$ = 1662 km², $A_{grid}$ = 1415 km² and $N$ = 1). The dot denotes**
**the lake centroid, respectively. The arrows indicate the flow direction, and the numbers indicate the grid cell levels. The blue-shaded grid cells represent those selected for the streamflow estimation, according to the workflow (Figure 2).**

However, for N > 1 (Approach I.b), the catchment occupied multiple grid cells. As a result, both grid cells partially occupied by the lake and upstream grid cells were counted. In addition to the grid cells of level 0 ($i$), the number of upstream grid cells, $j = N - i$, were counted by levels, selecting as many grid cells as indicated by $j$. This ensures that the total contributing area, represented by $i + j$ grid cells, approximates the scaled catchment size $N$. Once all level 1 grid cells have been counted, we proceed to the next level, continuing the process until $j$ grid cells have been counted (Figure 4). If not all grid cells at the same level can be counted (when the available grid cells at a given level exceed $j$ or the remaining $j$), those with the steepest slope were prioritized (Figure 2). For example, lake 11693 (Figure 4) has an $A_{catchment}$ of 11316 km², while the grid cell where the lake centroid is located has an $A_{grid}$ of 1220.18 km², resulting in $N$ = 9. Of the 9 grid cells counted, 2 corresponded to level 0
($i$ = 2), representing grid cells partially occupied by the lake. For the upstream grid cells ($j$ = 7), 2 of the 7 grid cells corresponded to level 1, 2 to level 2, and 3 to level 3.

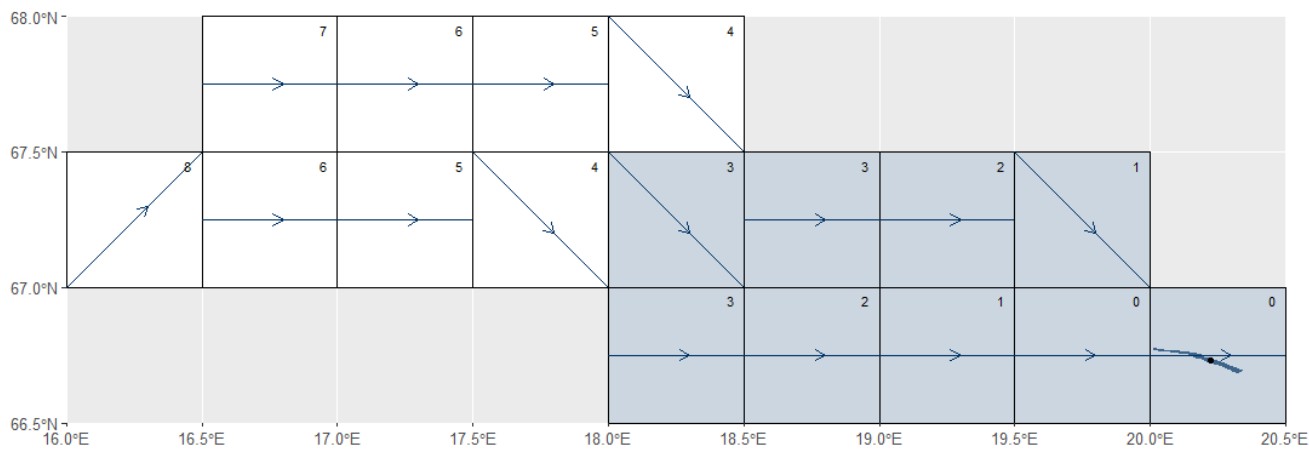

**Figure 4. Catchment-lake scheme for lake 11693 ($A_{lake}$ = 21.05 km², $A_{catchment}$ = 11316 km², $A_{grid}$ = 1220 km² and $N$ = 9). The dot denotes the lake centroid and the pour point, respectively. The arrows indicate the flow direction, and the numbers indicate the grid**
**cell levels. The blue-shaded grid cells represent those selected for the streamflow estimation, according to the workflow (Figure 2).**



For large lakes where the surface area, $A_{lake}$, exceeded the $A_{grid}$ (Approach II) the water flow from upstream grid cells was included as river discharge, $dis$ [m$^3$ s$^{-1}$], at the catchment grid cells bordering the lake grid cells ($k$, grid cells classified as level 1), in addition to $q_{tot}$ [m s$^{-1}$] and $q_g$ [m s$^{-1}$] proportional to the land area of the grid cells partially occupied by the lake ($i$, grid cells classified as level 0), $A_{catchment}^{grid}$ [m$^2$] (Figures 2 and 5). The inclusion of $dis$ was necessary because, in large lakes, a

significant portion of inflow enters not as diffuse runoff from adjacent land areas, but as a concentrated river discharge delivered through from upstream flow paths or tributaries. Lake grid cells ($i$) that did not flow into the lake, such as the grid cell where the outlet of the catchment is located or grid cells acting as sinks (usually to the ocean), were excluded. For example, lake Vänern – 105 (Figure 5) spans 12 grid cells ($i$) with an additional 18 upstream grid cells. Of the $i$ lake grid cells, the pour point grid cell was excluded because it was flowing out of the lake. Streamflow was calculated as the product of $q_{tot}+q_g$ and

$A_{catchment}^{grid}$ for each of the remaining 11 grid cells at level 0 ($i = 11$). Additionally, the $dis$ contribution from 7 bordering grid cells within the 18 upstream cells ($k = 7$, level 1 grid cells) was included.







**Figure 5. Catchment-lake scheme for lake Vänern – 105 ($A_{lake}$ = 5486 km², $A_{catchment}$ = 48421 km² and $A_{grid}$ = 1604 km²). The dot and cross denote the lake centroid and the pour point, respectively. The arrows indicate the flow direction, and the numbers indicate the grid cell levels. The blue-shaded grid cells represent those selected for the streamflow estimation, according to the workflow (Figure 2).**

## 2.5 Validation of streamflow at catchment scale

Historical simulations of daily river discharge of the hydrological model HYPE (Hydrological Predictions for the Environment; Lindström et al., 2010), which is used operationally and was developed by Swedish Meteorological and Hydrological Institute



(SMHI), are openly available for 35447 sub-catchments across Europe over a 30-year period (1981-2010) (Donnelly et al., 2016; https://hypeweb.smhi.se/explore-water/historical%20data/europe-time-series). The HYPE discharge simulations (hereafter referred to as the reference dataset) were used to evaluate the performance of the developed methodology for scaling streamflow from grid cells to catchment scale.

The HYPE model was forced with ERA5 reanalysis climate data (Donnelly et al., 2016), ensuring that the simulations provided

an independent dataset for validation. Daily HYPE outputs were averaged to produce monthly and annual discharge values, which were then compared to our monthly estimations and derived annual averages calculated from the monthly values over the common period (1981-2010) across 70 study sites.

Monthly and annual averages derived from daily observed river discharge at stations downstream of 6 large, medium and small-sized lakes, which are also representative lakes within the ISIMIP Global Lake Sector (lakes: Vänern – 105, Vättern –

104, Mälaren – 102, Siljan – 1150, Erken – 12809 and lake 149288), available from the Swedish Meteorological and Hydrology Agency (SMHI; https://www.smhi.se/data), were also compared with the corresponding monthly and annual average simulations. Note that streamflow simulations for lake 149288 were included in the validation against observations, but were not included in the validation against reference values. Although the observed data represent discharge downstream of the lakes (lake outflows), while the simulations estimate lake inflows, we assume than lake evaporation in Sweden is relatively

minor compared to total inflow and outflow volumes, particularly at monthly and annual timescales.

Performance was assessed using the Kling-Gupta efficiency, *KGE*, metric. *KGE* decomposes model performance into three aspects: the linear correlation coefficient, $KGE_r$, the bias ratio, $KGE_b$, and the variability ratio, $KGE_g$, which assess the model's ability to reproduce timing, mean and variability, respectively (Gupta et al., 2009; Kling et al., 2012).

$$KGE = 1 - \sqrt{(KGE_r - 1)^2 + (KGE_b - 1)^2 + (KGE_g - 1)^2} \tag{1}$$


$$KGE_b = \frac{\mu_{sim}}{\mu_{obs}} \tag{2}$$

$$KGE_g = \frac{CV_{sim}}{CV_{obs}} = \frac{\frac{\sigma_{sim}}{\mu_{sim}}}{\frac{\sigma_{obs}}{\mu_{obs}}} \tag{3}$$

where $\mu_{sim}$ and $\mu_{obs}$ are simulated and observed mean, and $\sigma_{sim}$ and $\sigma_{obs}$ are simulated and observed standard deviation. All three metrics have an optimum value of 1. The three individual metrics are combined into an overall model performance, *KGE*, by calculating the Euclidean distance from the ideal point. The error term is subtracted from unity to constrain the metric between

1 (perfect agreement) and –∞.

Additional goodness-of-fit metrics for comparing reference and simulated values, such as the Mean Bias Error (*MBE*), Root Mean Square Error (*RMSE*), Normalized Root Mean Square Error (*NRMSE*) and Nash-Sutcliffe Efficiency (*NSE*; Nash & Sutcliffe, 1970), can be found in the Supplementary Material.





## 3. Results

The performance of the scaled streamflow from grid cells to the catchment scale (hereafter referred to as simulations) for monthly time series over the period 1981-2010 across 70 study sites is shown in Figure 6A and Table S2. The average Kling-Gupta efficiency, $KGE$, was 0.59±0.18 (mean ± standard deviation), with values ranging from -0.07 to 0.86. For all study sites, the $KGE$ exceeded -0.41, indicating that the simulated streamflow provided added value compared to using long-term mean values. The average correlation coefficient, $KGE_r$, of 0.79±0.08 suggested a relatively satisfactory performance in terms of

monthly streamflow timing.

The bias measure, $KGE_b$, averaged 1.06±0.30, was close to the optimal value of 1. The variability ratio, $KGE_g$, averaged 0.88±0.22, indicating good performance but also reflecting some underestimation of variability in the simulations. In 52 out of the 70 study sites (74 %), $KGE_r$ was greater than 0.75, and $KGE_g$ ranged from 0.75 to 1.25. Meanwhile, in 39 out of the 70 study sites (56 %), $KGE_b$ ranged from 0.75 to 1.25. This suggests that while the overall volume of water flowing to the lakes

was on average well simulated (mean $KGE_b$ of 1.06), there was variability among the lakes in the accuracy of the simulated inflowing water volume (standard deviation of $KBG_b$ of 0.30). Accordingly, $KGE_b$ contributed more to lower overall KGE than problems related to simulating timing ($KGE_r$) or variability ($KGE_b$) (Figure 6A).

The inter-annual variability of streamflow was assessed by comparing the simulated and reference mean streamflow for each year (Table S2; Figure S2). The average values of the $KGE$ components ($KGE_r$ of 0.77±0.14, $KGE_b$ of 1.06±0.30, $KGE_g$ of

1.06±0.31) indicated an overall good performance in responding differently to wet and dry years. Combining the $KGE$ components resulted in an overall $KGE$ of 0.54±0.23, which is comparable to the $KGE$ (0.59±0.18) for monthly streamflow.

Of the 70 study sites where simulated and reference streamflow were compared, streamflow for 68 study sites was calculated following Approach I ($A_{lake} \leq A_{grid}$), and of these, 39 study sites were analysed according to Approach I.a ($N \leq 1$) and 29 study sites according to Approach I.b ($N > 1$). The remaining 2 study sites followed Approach II ($A_{lake} > A_{grid}$) – the large lakes

Vänern (105) and Vättern (104). When $A_{lake} \leq A_{grid}$, the overall average $KGE$ for simulated seasonal streamflow was similar for both Approaches I.a ($KGE$ of 0.56±0.15) and I.b ($KGE$ of 0.60±0.21).

In 5 out of 29 study sites using Approach I.b ($A_{lake} \leq A_{grid}$ for $N > 1$), only the grid cells with the steepest slope were counted at the last level, rather than all grid cells. Performance was calculated in both cases: counting only the grid cells with the steepest slope or counting all cells at the last level. In both cases, the performance was acceptable, and the differences between

$KGE$ and its components were marginal (counting all grid cells at the last level: $KGE$ of 0.49±0.31, with $KGE_r$ of 0.76±0.05, $KGE_b$ of 0.87±0.17, $KGE_g$ of 1.23±0.49; counting grid cells with the steeper slope: $KGE$ of 0.48±0.31, with $KGE_r$ of 0.74±0.07, $KGE_b$ of 0.85±0.15, $KGE_g$ of 1.22±0.49).

For Approach II ($A_{lake} > A_{grid}$), the performance was good in both Vänern (Figure 6) and Vättern, with a $KGE$ of 0.77 ($KGE_r$ of 0.85, $KGE_b$ of 0.97, $KGE_g$ of 1.17) and 0.79 ($KGE_r$ of 0.79, $KGE_b$ of 0.97, $KGE_g$ of 1.00), respectively.

Lake Mälaren (102), the third largest lake in Sweden, extends over 9 grid cells (Figure S1); however, its $A_{lake}$ (of 1083 km$^2$) does not exceed the $A_{grid}$ of 1,580 km$^2$ due to its irregular and branched shape. Scaling streamflow Approach I.b ($A_{lake} \leq A_{grid}$



for $N > 1$) and Approach II ($A_{lake} > A_{grid}$) were tested (Figure 2). For Approach I.b, simulated streamflow showed good performance at the seasonal scale, with $KGE$ of 0.71 ($KGE_r$ of 0.72, $KGE_b$ of 1.04, $KGE_g$ of 1.06); however, errors in reproducing the timing of flow decreased the overall performance. In Approach II, the simulated seasonal streamflow was less accurate, with a $KGE$ of 0.47 ($KGE_r$ of 0.52, $KGE_b$ of 0.98 and $KGE_g$ of 0.80). The errors were caused by either a reduced ability to accurately reproduce the timing of flow increases and decreases; and an underestimation of the magnitude of the variability, although it was still reasonably good.

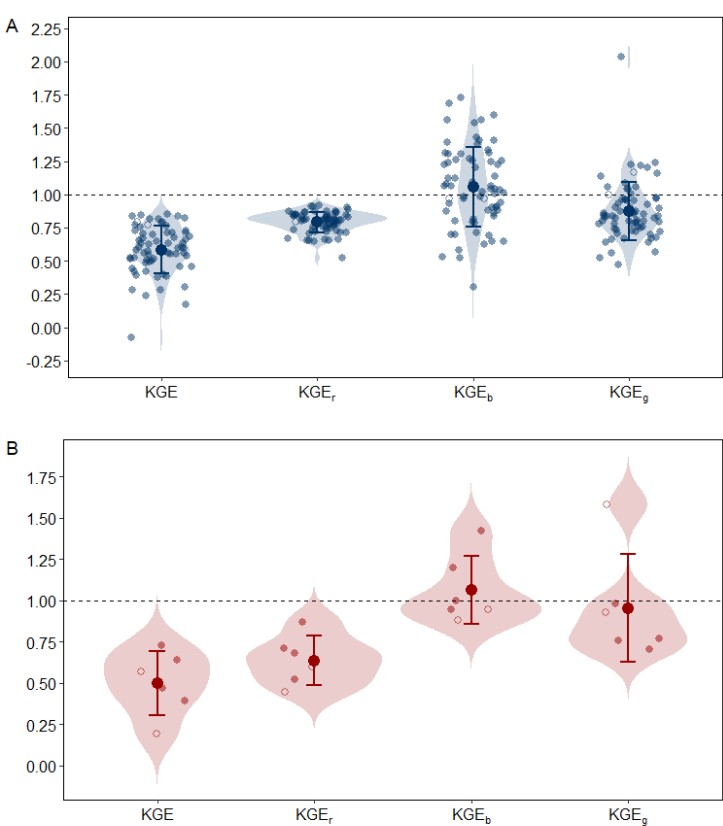

**Figure 6. Kling-Gupta efficiency ($KGE$) and its components, timing ($KGE_r$), bias ($KGE_b$) and variability ($KGE_g$), when comparing monthly simulations of scaled streamflow with reference data from the HYPE model (A) and observed data (B) of streamflow. Filled dots correspond to study sites where $A_{lake} \leq A_{grid}$ (approach I) and unfilled dots corresponds to study sites where $A_{lake} > A_{grid}$ (approach II). The horizontal dashed line marks where $KGE$ and its components equal 1, representing a perfect match**

In addition, the performance of simulated streamflow was assessed by comparing simulations with observations for 6 study sites, which are both representative lakes in the ISMIP3 Global Lake Sector and for where observations are available (Figure 6B; Table S3). At the seasonal scale, the average $KGE$ was 0.50±0.19, with $KGE_r$ of 0.64±0.15, $KGE_b$ of 1.06±0.21, $KGE_g$ of 0.96±0.33. The overall performance was reduced due to the timing of the flow. At the annual scale, the performance of the scaling streamflow from grid cells to catchment scale was good ($KGE$ of 0.73±0.16, with $KGE_r$ of 0.83±0.05, $KGE_b$ of 1.07±0.21, $KGE_g$ of 0.97±0.16), both in overall and in terms of timing, bias and variability (Figure S2).



We conclude that the overall performance of the scaled streamflow simulations matched satisfactorily to both reference

(derived from the hydrological model HYPE) and observed streamflow (Figure 7; Figures S3-S7).

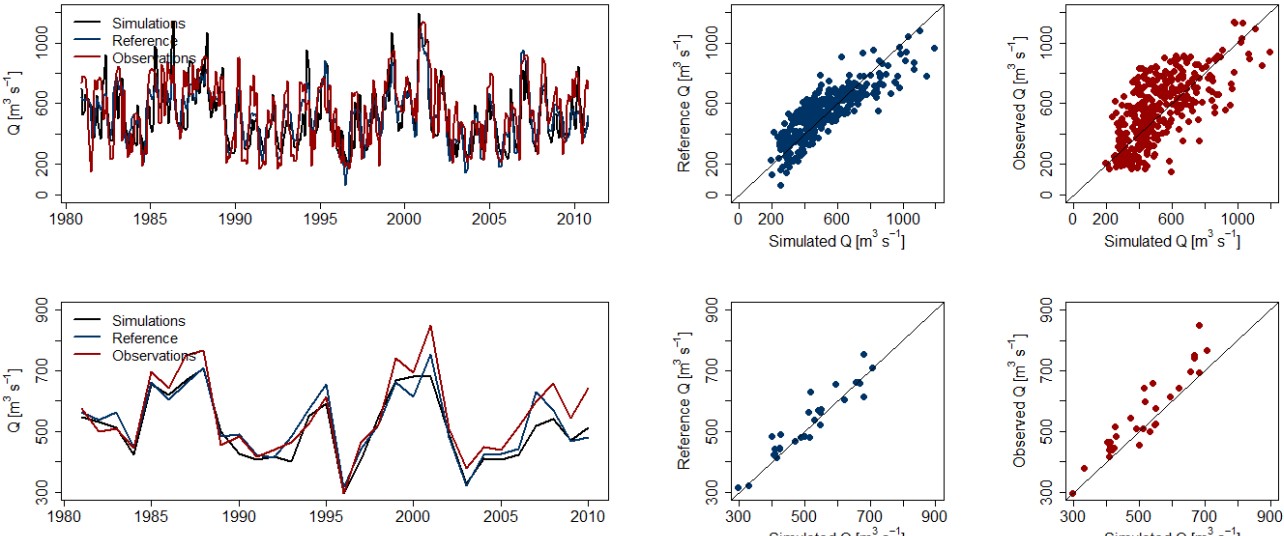

**Figure 7. Monthly (upper row) and annual (lower row) comparison of simulated (black), reference (blue) and observed (red) streamflow for lake Vänern (105) over a 30-year period (1981–2010).**

## 4. Discussion

This study demonstrates a scaling approach that reliably estimates streamflow to individual lakes from gridded streamflow data, showing strong agreement between simulations, reference data, and observations. The fit of our simulated scaled streamflow was acceptable regardless of temporal scale (monthly, Figure 6; annually, Figure S2), and geospatial configuration i.e. the location and size of the lake and its catchment, corresponding to the scaling Approaches I.a, I.b and II (Figure 6; Figure S2). The simulated scaled streamflow also fitted reference data as well as observations equally well (Figures 6-7; Figures S3-

S7). These conclusions are based on the evaluation of the performance of scaled streamflow simulations across 71 study sites in Sweden, and the country's diverse landscape, which includes a wide range of lake sizes, catchment sizes and land covers, climate conditions and topography, thus providing a robust basis for assessing model performance. This diversity enhances the generalizability of the validation results across different hydrological settings. Further, Donnelly et al. (2016) demonstrated that Sweden has a well-established hydrological modeling framework, particularly through the HYPE (Hydrological

Predictions for the Environment) model, which has been extensively applied and validated in the region, and which was used as a reference data source in this analysis. In addition, Sweden has long-term, high-resolution datasets for precipitation, temperature and streamflow, which are essential for ensuring robust hydrological simulations, making the HYPE model optimal for testing the accuracy of scaled streamflow simulations. We feel that the regionally focused nature of HYPE should provide an excellent comparative data set to the globally applied WaterGAP 2 model. Comparisons to HYPE were used to





judge our scaling approach, since HYPE simulations are readily available for the catchment associated with lake inflows, while measured lake inflow data are far less common.

While this study focuses exclusively on Swedish lakes, the wide range of topographic and geomorphological conditions represented in the datasets supports the potential global applicability of the scaling approach (Table S1). The dataset spans more than three orders of magnitude in lake surface area (from 0.34 to 5487 km$^2$) and are embedded in catchments ranging

from 101 to 48421 km$^2$, with catchment-to-lake area ratios varying from 3.37 to 14962. These systems span a broad latitudinal gradient, from approximately 55°N to 69°N, encompassing temperate subarctic climates, and also cover a wide elevational range, from lowland lakes near sea level to high altitudes systems. This introduces variability in temperature regimes, snow accumulation and runoff dynamics. Catchment topography is similarly diverse, with mean catchment slopes ranging from 0.000001 to 0.012 m m$^{-1}$, and a wide spread in both minimum and maximum catchment slopes that influence flow

concentration and hydrologic connectivity. This richness in latitude, elevation, slope, area and catchment configuration reflects a broad spectrum of geomorphic and hydrological settings. Since these land-based drivers are primary controls on surface hydrology and lake water balances, their strong representation in the Swedish dataset supports the scaling approach's transferability to other regions. While the Swedish climate is temperate to subarctic, factors such as evaporation may differ in arid and tropical conditions. Thus, although climate-related refinements may be necessary for certain regions, the core method

grounded in topographic and geometric scaling is broadly applicable.

Although our scaling approach is effective for estimating how much water flows into lakes, it does not account for the full routing of water through rivers. A key limitation comes from differences in the spatial detail of the datasets we used. Water fluxes ($q_{tot}$, $q_g$ and $dis$) are provided by WaterGAP 2.2e on a coarse grid ($0.5°$ grid cell), while geometry of lake boundaries in the HydroLAKES dataset is represented at a much finer scale. This mismatch in resolution can lead to inconsistencies, for

example: an inflowing stream might appear to flow into a lake in the gridded data, even though in reality it joins the river downstream. These issues are especially common in small lake systems. To reduce their impact, we used known catchment areas to adjust our streamflow estimates and avoid large overestimations. However, this approach does not fully resolve the mismatches, and it breaks the water mass balance, meaning we may misrepresent how much water flows through the system. If the objective is to model the transport of water through river and lake networks, additional considerations would be required

to ensure an accurate mass balance.

The task of linking ISIMIP Global Water Sector and Lake Sector models requires the use of gridded models and gridded data. One critical factor influencing the accuracy of the simulated streamflow is the structure and potential limitations of the gridded hydrological model. In this study, the gridded water flux ($q_{tot}$, $q_g$ and $dis$) simulations were obtained from WaterGAP 2.2e (Müller Schmied et al., 2023), a global hydrological model that simulates water availability and use at a global scale. The

performance of the scaling approach, when the simulated streamflow is compared to reference values or observations, is thus inherently linked to the accuracy of WaterGAP 2.2e outputs. The WaterGAP 2.2e model achieved a global monthly streamflow performance with a median *KGE* of 0.58 (Müller Schmied et al., 2023). The scaled streamflow simulations performed similarly, with a median *KGE* of 0.59 compared to the reference and a median *KGE* of 0.52 against observations. Regarding





$KGE$ components, the bias ratio ($KGE_b$) showed a median value close to the optimal value of 1 in all cases (for WaterGAP

2.2e: median $KGE_b$ of 1.01 and for scaled streamflow: median $KGE_b$ of 1.04 compared to the reference and median $KGE_b$ of
0.98 compared to observations). $KGE_g$ deviates from 1, indicating that streamflow variability is not well simulated, the median
$KGE_g$ was 0.86 for WaterGAP 2.2e, while the scaled streamflow showed median $KGE_g$ values of 0.84 compared to the
reference and 0.85 compared to observations. This underestimation of streamflow variability suggests that hydrological
extremes, including peak and low flows, may not be fully captured when using a gridded model with gridded meteorological

forcing, potentially leading to a smoothing effect in the simulations. In the Köppen-Geiger climate region D, which includes
Sweden, 47 % of the gauging stations used for the calibration of WaterGAP 2.2e showed $KGE_g$ values between 0.5 and 0.9.
For the scaled streamflow, 61 % of the study sites fell within 0.5-0.9 range when compared to the reference, while 43 % did
so when compared to observations. The ability to capture the timing of streamflow increases and decreases was generally good
in WaterGAP 2.2e, with a median $KGE_r$ of 0.78. In the Köppen-Geiger climate region D, 27 % of gauging stations showed a

$KGE_r$ below 0.5, indicating moderate timing errors. The scaled streamflow showed improved timing accuracy when compared
to the reference, with a median $KGE_r$ of 0.81 and only 11% below 0.5. However, when compared to observations, the median
$KGE_r$ decreases to 0.64, with 50% of study below 0.5, suggesting that while the scaling approach improves consistency with
the reference, discrepancies remain when validated against observed streamflow.

Another factor influencing the performance of streamflow simulations was the data source used for validation. E-HYPE, the

European-scale implementation of the HYPE model, utilized gauged streamflow data for catchments larger than 5000 km$^2$,
while for smaller catchments, it relies on modelled ungauged streamflow (Donnelly et al., 2016). This distinction is important
because it might explain why the accuracy of streamflow simulations tended to be higher in well-gauged large catchments,
whereas streamflow simulations in smaller catchments were inherently more uncertain due to the reliance on hydrological
modeling rather than observations. Among the 70 study sites, 23 had a catchment area larger than 5000 km$^2$, yielding a mean

$KGE$ of 0.60±0.22, while the remaining 47 study sites ($\leq$ 5000 km$^2$) exhibited a comparable performance, with a mean $KGE$
of 0.58±0.15. However, for the 6 study sites where scaled streamflow was compared directly with observations, performance
varied: 4 study sites with catchments larger than 5000 km$^2$ showed a mean $KGE$ of 0.47±0.20, whereas another 2 study sites
($\leq$ 5000 km$^2$) achieved a mean $KGE$ of 0.56±0.24. This suggests that while scaled streamflow simulations performed similarly
across different catchment sizes when compared to a streamflow reference, their accuracy was more variable when directly

validated against observations.

It is important to point out the difference between the reference (E-HYPE discharge simulations) and scaled streamflow are
due not only to potential inaccuracies in the scaling approach itself, but also to differences in the performance of E-HYPE and
WaterGAP 2.2e models, and meteorological data used to force both models. The good agreement between the reference and
simulated scaled streamflow, despite multiple potential sources of errors, therefore, suggests that the scaling approach

presented here is likely to perform similarly well in another region of the world.

When scaling streamflow from gridded data to the catchment scale, three different approaches (Approach I.a, Approach I.b
and Approach II) were employed to account for differences in the size and location of lakes and their catchments in relation to





grid cells. These approaches are essential due to the significant differences in lake size, shape and catchment area relative to the model grid resolution. The performance of the scaling approach was further assessed across different catchment and lake sizes. For lakes where $A_{lake} \leq A_{grid}$, both sub-approaches (I.a and I.b) yielded similar results, demonstrating the robustness of the method across different grid configurations. In large lakes where $A_{lake} > A_{grid}$ (Approach II), such as Vänern and Vättern, the performance was strong with $KGE$ values of 0.77 and 0.79, respectively. In contrast, for Lake Mälaren, which has a highly irregular shape (Figure S1), the choice of scaling approach significantly affected performance. The better performance of Approach I.b ($KGE$=0.71) compared to Approach II ($KGE$=0.47) highlights the importance of accounting for complex lake morphologies in streamflow scaling. This suggests that for lakes with complex morphologies, different scaling approaches should be tested to determine the most suitable method. As demonstrated in the case of Lake Mälaren, where performance varied notably between approaches, careful evaluation of lake shape and catchment characteristics is crucial to achieving accurate streamflow scaling

In addition, the validation against observed streamflow data for six representative lakes (Figure 6B; Table S3) confirmed the ability of the scaled simulations to match not only reference data, but also observed data. Seasonal-scale performance was slightly lower ($KGE$ of 0.50±0.19) due to timing errors, compared to stronger annual-scale performance ($KGE$ of 0.73±0.16), indicating that the method effectively captures long-term hydrological trends.

## 5. Conclusion

The results of this study demonstrate that the developed scaling approach is reliable and robust for global applications, showing good performance across a wide range of hydrological settings. By implementing three distinct approaches (I.a, I.b and II), the methodology effectively accounts for varying lake sizes and catchment configurations, from small single-grid lakes and catchments to large, complex multi-grid systems such as Vänern, Vättern and Mälaren. This flexibility enables consistent application across diverse hydrological regimes and supports the use of the method in a large-scale modelling framework. While the overall performance was satisfactory, evidence by strong average $KGE$ values, some limitations remain, particularly in capturing flow variability and timing in more complex systems.

This study also addresses a key limitation in the ISIMIP framework by introducing a method to dynamically link gridded catchment hydrology with lake inflows. The approach allows lake simulations to reflect not only direct climate impacts but also changes in upstream hydrological processes, enabling more realistic assessments of climate change impacts on lakes globally. By bridging the gap between catchment and lake dynamics, this methodology provides a valuable tool for improving integrated hydrological and biogeochemical lake modelling within ISIMIP and beyond.



**Acknowledgements**

We are grateful to ISIMIP for producing, coordinating and making available the ISIMIP experiments. We also acknowledge the contribution of the ISIMIP Global Water Sector modelling teams for providing the hydrological simulations that formed the basis of this study. This research funding was supported by Vetenskapsrådet under project ID 2021-04639.

**Code availability**

All R scripts produced during this study are available at https://doi.org/10.5281/zenodo.15925096 (Ayala, 2025a).

**Data availability**

The impact model WaterGAP 2.2e simulations of monthly discharge (*dis*), total runoff ($q_{tot}$) and groundwater runoff ($q_g$) (Müller Schmied et al., 2024b) for the standard evaluation experiment *obsclim_histsoc_default* (Frieler et al., 2024), along

with the drainage direction map and slopes for river routine, are available in the ISIMIP repository (https://data.isimip.org) and at https://doi.org/10.5281/zenodo.15917845 (Ayala, 2025b). Representative lakes at the ISIMIP3 Global Lake Sector can be accessed at https://github.com/icra/ISIMIP_Lake_Sector. Scaled streamflow simulations for the 71 studied sites are available at https://doi.org/10.5281/zenodo.15919494 (Ayala, 2025c).

**Author contributions**

AIA lead the study. AIA and JLH developed the scaled method with contributions from SNG, DCP and SS. AIA and JLH generated and validated the scaled streamflow data. AIA wrote the paper with contributions from JLH, DMB, RM, SNG, DCP and SS.

**Competing interests**

The contact author has declared that neither of the authors has any competing interests.

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
