# Peer review of "Integration of the Global Water and Lake Sectors within the ISIMIP framework through scaling of streamflow inputs to lakes"

_EGUsphere, 2025_

## Author Comment (AC1)

Ayala, A. I., Hinostroza, J. L., Mercado-Bettín, D., Marcé, R., Gosling, S. N., Pierson, D. C., and Sobek, S.: Integration of the Global Water and Lake Sectors within the ISIMIP framework through scaling of streamflow inputs to lakes, EGUsphere [preprint], https://doi.org/10.5194/egusphere-2025-3126, 2025.

**Reviewer 1 (Miaohua Mao)**

**Summary:**

This work integrates the stream flows from the nearby catchments into 71 lakes in Sweden, based on the scaling method of the global water and lake sector model. The model performances are compared with referenced model results and observed data from stations. The authors finally conclude that the updated model is satisfactory on modeling the streamflow. The authors have done a good work in explaining the workflow of their coded work, while the reviewers have some comments and suggestions needed to be clarified before it can be published after Minor Revision.

We thank the reviewer for the constructive comments and thoughtful suggestions, which have helped us improve the quality and clarity of the manuscript. Below, we provide a detailed response to each comment, and we indicate how the manuscript has been revised accordingly.

**Comments:**

**Comment 1:**

The reviewer's suggestion is avoiding using the specific values for the KGE in this Abstract section. Instead, this section should provide epitome of the entire work in a succinct and clear way.

**Reply:**

We have revised the Abstract by removing the specific *KGE* values rephrasing the content to provide a more general summary of main findings of the study. The part containing the *KGE* values was replaced by the following revised sentences: "The methodology was applied to 70 lakes across Sweden covering a wide range of sizes, hydrological settings and catchment characteristics. The estimated streamflow was validated against both the streamflow outputs from the hydrological model HYPE and observed data. The comparison demonstrated good agreement in terms of long-term streamflow mean and seasonal pattern, indicating that the proposed approach is capable of producing reliable streamflow estimates without requiring high-resolution local models."

**Comment 2:**

The authors have done a good work in introducing the previous study work and its research gap, and what they need to do to fill this research gap, i.e., develop the coupled streamflow and lake model via the various discharges (e.g., surface, subsurface, groundwater etc.)

We thank the reviewer for this positive feedback. We are pleased that the research gap and motivation of our study were clear and well received.

**Comment 3:**

Material and methods. This section is generally well written and Fig. 2, 3, and 4 are nice figures to illustrate the procedure of the modeling frame well. Regarding the Section 2.5 Validation of streamflow at catchment scale, it is better by providing the range for the quality of Kling-Gupta efficiency (KGE) values. For example, in which ranges stand for model performance is excellent, good, poor etc., and this definition needs some references to support it. Another

comment is to define the CVsim and CVobs, which the reviewer considers as Coefficient of Variation.

Reply:

We appreciate the reviewer's positive feedback on the Material and Methods section and the figures illustrating the modeling framework.

Regarding the suggestion to provide interpretation range for the Kling-Gupta Efficiency (KGE), we ha now included the following classification, based on Knoben et al. (2019):

| KGE =1 | Perfect    |
|---------------|------------|
| 0.75≤KGE<1    | Very good  |
| 0.5≤KGE<0.75  | Good       |
| 0.25≤KGE<0.5  | Acceptable |
| KGE<0.25      | Poor       |

This classification has been added to Section 2.5 of the manuscript, along with the appropriate reference.

In addition, we clarify that the KGE was calculated using the KGE() function from the hydroGOF R package, with method="2012" to follow the revised formulation proposed by Kling et al. (2012). In this version, variability is represented by  $KGE_g$ , defined as the ratio of the coefficient of variation of the simulated values to the observed values:  $KGE = \frac{CV_{sim}}{CV_{obs}} = \frac{\sigma_{sim}/\mu_{sim}}{\sigma_{obs}/\mu_{obs}}$

$$KGE = \frac{CV_{sim}}{CV_{obs}} = \frac{\sigma_{sim}/\mu_{sim}}{\sigma_{obs}/\mu_{obs}}$$

where  $\sigma$  and  $\mu$  denote the standard deviation and mean of the simulated and observed time series respectively.

We also note that the definition of the coefficient of variation (CV) is already provided in Equation 3 of section 2.5.

The *KGE* classification has been incorporated into the Material and Methods section as follows: "Based on Knoben et al. (2019), KGE is interpreted as: KGE=1 perfect agreement, 0.75\leq KGE\leq 1 very good performance, 0.50\leq KGE\leq 0.75 good performance, 0.25\leq KGE\leq 0.50 acceptable performance and KGE<0.25 poor performance."

**References:**

Kling, H., Fuchs, M., and Paulin, M.: Runoff conditions in the upper Danube basin under an ensemble of climate change scenarios, J. Hydrol., 424-425, 264-277, https://doi.org/10.1016/j.jhydrol.2012.01.011, 2012.

Knoben, W. J. M., Freer, J. E., and Woods, R. A.: Technical note: Inherent benchmark or not? Comparing Nash-Sutcliffe and Kling-Gupta efficiency scores, Hydrol. Earth Syst. Sci., 23, 4323–4331, https://doi.org/10.5194/hess-23-4323-2019, 2019.

**Comment 4:**

Line 229: '..... we assume than lake evaporation .....' maybe changed to '..... we assume that lake evaporation ......'

Reply:

Thank you for noticing this typographical error. It has been corrected as suggested in the revised manuscript.

**Comment 5:**

Line 223 and other places: The authors please make sure that whether 70 or 71 lakes in Sweden are studied. This needs to be consistent throughout the texts.

**Reply:**

A total of 71 lakes in Sweden were initially considered. However, streamflow simulations for lake 149288 were included in the validation against observations, but not in the validation against reference values due to data limitations. This discrepancy caused some confusion in the lake count in the different sections of the text. We have now removed lake 149288 from the analysis entirely, as it was only partially included in the original validation. As a result, the total number of lakes studied is now 70. We have carefully reviewed and revised the manuscript to ensure that this number is consistent throughout the text and have removed any reference to the previously included lake 149288.

**Comment 6:**

Line 248-249: 'For all study sites, the KGE exceeded -0.41, indicating that the simulated streamflow provided added value compared to using long-term mean values.' The reviewer is a little bit confused that a negative value of KGE (e.g., -0.41) means this revision provides added value.

**Reply:**

Negative *KGE* values generally indicate poor model performance, the original intent of the sentence was to highlight that the model still performed better than a simple baseline (e.g. using the mean observed discharge as prediction).

To avoid confusion, we have revised the sentence for clarity: "For all study sites, the *KGE* exceeded -0.41, indicating that the simulated streamflow provided added value compared to simple prediction based on the long-term mean streamflow"

**Comment 7:**

In the Results Section. The authors have compared their model with reference results and observed data, respectively. The reviewer considers that would that also necessary to compare the reference results with the observed data. By doing so, the reader could have better understanding that whether the developed model improves its performance or not, compared with the traditional model (i.e., the reference results).

Reply:

Lake 149288, which had streamflow observations available but lacked reference data, was excluded from the original validation analysis to maintain consistency throughout the text. To ensure comprehensive comparison, additional lakes with both streamflow observations and reference data were included in the analysis. As a result, streamflow simulations were compared with observations for a total of 10 lakes.

Additionally, in the revised results sections, we have incorporated a comparison between the reference and observed streamflow, using the same performance metrics. Note that, while observations were available for 10 lakes, comparison were conducted for only 9 lakes, as reference streamflow data are available from 1981-2010. A new table (Table S4) presenting performance metrics has been added to the supplementary material and a brief description of the performance has been added to the result section.

Brief description of the performance added to the result section: "Finally, a further evaluation was conducted by comparing reference and observed streamflow for 9 study sites (note that the reference and observations datasets cover different time periods, which limited direct comparability in the 10 study sites for which observations were available) (Table S4). At the monthly scale, the average KGE was  $0.44\pm0.44$  (with  $KGE_r$  of  $0.65\pm0.23$ ,  $KGE_b$  of  $1.12\pm0.34$ ,  $KGE_g$  of  $1.13\pm0.46$ ), indicating on average acceptable agreement with substantial inter-site differences. At the yearly scale, performance improved to KGE of  $0.55\pm0.26$  (with  $KGE_r$  of  $0.78\pm0.12$ ,  $KGE_b$  of  $1.12\pm0.34$ ,  $KGE_g$  of  $0.77\pm0.19$ ). Overall, these results demonstrate that the

| scaling method provides added value, improving the simulations of streamflow compared with standard catchment-scale hydrological models." |
|-------------------------------------------------------------------------------------------------------------------------------------------|
|                                                                                                                                           |
|                                                                                                                                           |
|                                                                                                                                           |
|                                                                                                                                           |
|                                                                                                                                           |
|                                                                                                                                           |
|                                                                                                                                           |
|                                                                                                                                           |
|                                                                                                                                           |
|                                                                                                                                           |

Table S4 of supplementary material:

| Lake     | Name        | Latitude | Longitude | MBE    | RMSE | NRMSE | NSE   | KGE   | $KGE_r$ | $KGE_b$ | $KGE_g$ | Frequency |
|----------|-------------|----------|-----------|--------|-------------|-------|-------|-------|---------|---------|---------|-----------|
| 102      | Mälaren     | 59.49    | 16.79     | 31.66  | 119.55      | 0.16  | 0.24  | 0.56  | 0.64    | 1.19    | 0.83    | monthly   |
| 102      | Mälaren     | 59.49    | 16.79     | 32.17  | 38.46       | 0.22  | 0.13  | 0.73  | 0.87    | 1.19    | 0.86    | yearly    |
| 104      | Vättern     | 58.33    | 14.49     | 3.03   | 25.64       | 0.31  | -1.25 | 0.24  | 0.39    | 1.07    | 1.45    | monthly   |
| 104      | Vättern     | 58.33    | 14.49     | 2.94   | 9.43        | 0.22  | 0.33  | 0.47  | 0.63    | 1.07    | 0.64    | yearly    |
| 105      | Vänern      | 58.88    | 13.55     | -25.90 | 136.73      | 0.14  | 0.60  | 0.71  | 0.79    | 0.95    | 0.81    | monthly   |
| 105      | Vänern      | 58.88    | 13.55     | -25.32 | 57.00       | 0.10  | 0.80  | 0.86  | 0.92    | 0.95    | 0.89    | yearly    |
| 12423    |             | 62.05    | 14.15     | -51.94 | 77.45       | 0.23  | -1.20 | -0.34 | 0.53    | 0.57    | 2.18    | monthly   |
| 12423    |             | 62.05    | 14.15     | -51.70 | 53.28       | 0.55  | -4.30 | 0.56  | 0.90    | 0.57    | 0.97    | yearly    |
| 12791    |             | 60.07    | 15.57     | 4.17   | 17.94       | 0.17  | -0.53 | 0.48  | 0.65    | 1.19    | 1.33    | monthly   |
| 12791    |             | 60.07    | 15.57     | 4.24   | 5.76        | 0.32  | -0.13 | 0.60  | 0.72    | 1.20    | 0.78    | yearly    |
| 12809    | Erken       | 59.84    | 18.60     | 0.69   | 1.27        | 0.40  | -2.72 | -0.14 | 0.27    | 1.86    | 0.84    | monthly   |
| 12809    | Erken       | 59.84    | 18.60     | 0.70   | 0.73        | 0.57  | -5.03 | -0.01 | 0.75    | 1.87    | 0.55    | yearly    |
| 12965    | Roxen       | 58.49    | 15.63     | 14.20  | 31.24       | 0.20  | 0.41  | 0.62  | 0.75    | 1.18    | 0.78    | monthly   |
| 12965    | Roxen       | 58.49    | 15.63     | 14.86  | 22.19       | 0.29  | 0.08  | 0.35  | 0.74    | 1.18    | 0.43    | yearly    |
| 142240   |             | 66.66    | 22.22     | 1.20   | 7.08        | 0.08  | 0.85  | 0.88  | 0.93    | 1.08    | 0.95    | monthly   |
| 142240   |             | 66.66    | 22.22     | 1.18   | 3.07        | 0.21  | -0.02 | 0.59  | 0.60    | 1.08    | 0.99    | yearly    |
| 152977 H | Hasselasjön | 62.08    | 16.78     | -0.31  | 2.78        | 0.06  | 0.88  | 0.92  | 0.94    | 0.96    | 1.04    | monthly   |
| 152977 H | Hasselasjön | 62.08    | 16.78     | -0.18  | 1.10        | 0.11  | 0.78  | 0.79  | 0.89    | 0.98    | 0.82    | yearly    |

**Comment 8:**

The current writing of this part could be improved in a more detailed way. For example, providing some skill metric values that are specified (e.g., various KGE values), so that this work could be better summarized in a more strict way.

Reply:

We have revised the Results section to improve the clarity and structure of the performance evaluation. These changes make the summary more quantitative and structured, as suggested, and improve the overall readability of the results. The changes can be found throughout the revised results section. Furthermore, as show in the Supplementary Material (Tables S2-S4) additional performance metrics including *MBE*, *RMSE*, *NRMSE* and *NSE* are already provided for each study site. We believe these revisions address your concerns and improve the overall presentation of the results.

**Revised result section:**

[revised manuscript text omitted]

---

## Author Comment (AC2)

Ayala, A. I., Hinostroza, J. L., Mercado-Bettín, D., Marcé, R., Gosling, S. N., Pierson, D. C., and Sobek, S.: Integration of the Global Water and Lake Sectors within the ISIMIP framework through scaling of streamflow inputs to lakes, EGUsphere [preprint], https://doi.org/10.5194/egusphere-2025-3126, 2025.

**Reviewer 2**

**Summary:**

The study presents a method of re-scaling gridded water flow data on lake catchments. The method is developed within the framework of ISIMIP, the model intercomparison platform facilitating access to climate scenarios, models, and observational data for model validation. The manuscript is well-structured, clearly written, and addresses an important gap in coupling water flow and lake models on global scales. The proposed method uses a straightforward rescaling algorithm, differentiating between three options---the catchment is smaller than a single grid cell, the catchment is larger than, but the lake is smaller than a grid cell, and the lake is larger than a single grid cell. The approach has been validated against the long-term outputs of the operational regional hydrological model HYPE applied to 71 Swedish lakes and against a smaller observational dataset, demonstrating satisfactory performance. The results, summarized in two pages and two figures, are clear and concise. The impact on the modeling community can be however limited: while Swedish lakes provide a robust and diverse test case, the extrapolation to global conditions (particularly arid and tropical systems with highly variable evaporation and different hydrological regimes) remains speculative. Still, it is a valuable methodological contribution, with openly available code and datasets, which ensures reproducibility, and an initial step towards coupling lake and water flow modeling in climate models.

We thank the reviewer for the positive and constructive feedback. Regarding the concern about extrapolation to global conditions, we would like to clarify that our study focuses on rescaling streamflow inputs into lakes. All hydrological calculations are taken from the global hydrological model WaterGAP 2, which has been extensively validated a cross a range of climatic and hydrological regimes, including arid and tropical systems and thus encompassing variable hydrological and evaporation regimes. Our method does not perform new hydrological modeling but operates on the existing generated by WaterGAP 2, with the purpose to scale them to lake catchments. Therefore, the applicability of our approach globally relies on the underlying WaterGAP 2 outputs, not on the rescaling approach itself. The validation of our scaling approach was conducted on a wide variety of lake and catchment properties, particularly in terms of size, suggesting its suitability for global application.

The following sentences were removed from the Discussion section to avoid confusion: "While the Swedish climate is temperate to subarctic, factors such as evaporation may differ in arid and tropical conditions. Thus, although climate-related refinements may be necessary for certain regions, the core method grounded in topographic and geometric scaling is broadly applicable."

**Comments:**

**Comment 1:**

The case of Lake Mälaren demonstrates that irregular morphologies can strongly affect scaling performance. The authors might consider providing more concrete recommendations for how to approach such cases practically.

**Reply:**

The case of Lake Mälaren indeed highlights the impact of irregular morphologies on scaling performance. However, despite the lake's very complex shape and bathymetry, the modeling results were still satisfactory. Specifically, for Lake Mälaren, Approach I.b yielded a good performance with a *KGE* of 0.71, while Approach II showed acceptable performance with a *KGE* of 0.47. These results demonstrate that even in lakes with complex morphologies, both approaches can deliver at least acceptable performance.

Moreover, when comparing these results to other lakes (Manuscript: Figure 6 and Table S2), Lake Mälaren is not an outlier. Several other lakes with less complex shapes showed similar performance metrics, indicating that while morphology can influence predictive performance, it is not the sole determinant of success. This suggests that practical application of the scaling approaches remains viable even in morphologically complex systems.

The Discussion section has been revised to reflect these points: "In contrast, for Lake Mälaren, which has a highly irregular shape (Figure S1), the choice of scaling approach significantly affected performance. The better performance of Approach I.b (*KGE*=0.71) compared to Approach II (*KGE*=0.47) highlights the importance of accounting for complex lake morphologies in streamflow scaling. Nevertheless, both scaling approaches achieved satisfactory performance comparable to other lakes with less complex morphologies, indicates that, although lake morphology can influence performance, it is not the sole determining factor, further supporting the robustness and practical applicability of the scaling approaches even for lakes with complex morphologies."

**Comment 2:**

Only six lakes are compared against observed streamflow. While this is understandable due to data availability, a short description of the lakes representativity, in terms of lake size, geographical location, hydrological regime, would strengthen confidence.

Reply:

The observed streamflow records were extended to 10 lakes, which represent a diverse range of physical and hydrological characteristics. Geographically, these lakes are distributed across latitudes from  $58.33^{\circ}$  to  $66.66^{\circ}$ , covering southern, central and northern regions of Sweden (Table 1). The lake area spans three orders of magnitude from  $7.68 \text{ km}^2$  (lake 142240) to  $5486.23 \text{ km}^2$  (lake Vänern), with catchment areas that vary independently of lake size ( $A_{catchment}$  raged from  $138.70 \text{ km}^2$  to  $48421 \text{ km}^2$ ). This includes both small lakes with small catchments ( $A_{catchment} A_{lake}^{-1}$  of 5.99 - 1 lake Erken) and large catchments ( $A_{catchment} A_{lake}^{-1}$  of 139.91 - 1 lake Roxen), as well as large lakes with small catchments ( $A_{catchment} A_{lake}^{-1}$  of 3.37 - 1 Lake Vättern) and large catchments ( $A_{catchment} A_{lake}^{-1}$  of 139.91 - 1 lake hydrological characteristics of the study. Overall, despite the limited availability of observed streamflow data, these ten lakes provide a representative cross-section of the variability in lake size, catchment characteristics and geographical distribution within the study area.

Table 1: Characteristics of the study sites with available streamflow observations.

| Lake  | Name    | Longitude | Latitude | $A_{lake}$ [km 2 ] | Acatchment [km 2 ] | Acatchment Alake -1 |
|-------|---------|-----------|----------|-------------------------------|-------------------------------|--------------------------------|
| 102   | Mälaren | 16.79     | 59.49    | 1083.13                       | 22682.20                      | 20.94                          |
| 104   | Vättern | 14.49     | 58.33    | 1888.04                       | 6369.10                       | 3.37                           |
| 105   | Vänern  | 13.55     | 58.88    | 5486.23                       | 48421.00                      | 8.83                           |
| 1150  | Siljan  | 14.77     | 60.86    | 290.88                        | 12084.50                      | 41.54                          |
| 12423 |         | 14.15     | 62.05    | 63.59                         | 8357.00                       | 131.42                         |

| 12791              |       | 15.57 | 60.07 | 34.77 | 2213.30  | 63.66  |
|--------------------|-------|-------|-------|-------|----------|--------|
| 12809              | Erken | 18.60 | 59.84 | 23.14 | 138.70   | 5.99   |
| 12965              | Roxen | 15.63 | 58.49 | 94.55 | 13228.50 | 139.91 |
| 142240             |       | 22.22 | 66.66 | 7.68  | 1272.30  | 165.66 |
| 152977 Hasselasjön |       | 16.78 | 62.08 | 8.36  | 610.00   | 72.97  |

The Discussion section has been revised to reflect these points: "Although validation against observed streamflow is constrained due to data availability, the 10 lakes used for validation are broadly representative of the 70 lakes included in the study. Geographically, these lakes are distributed across latitudes from 58.33° to 66.66°, covering southern, central and northern regions of Sweden (Table S3). The lake area spans three order of magnitude from 7.68 km2 (lake 142240) to 5486 km2 (lake Vänern), with catchment areas that vary independently of lake size (Acatchment raged from 138.7 km2 to 48421 km2). This includes both small lakes with small catchments ( $A_{catchment} A_{lake}^{-1}$  of 5.99 – lake Erken) and large catchments ( $A_{catchment} A_{lake}^{-1}$ of 139.91– lake Roxen), as well as large lakes with small catchments ( $A_{catchment} A_{lake}^{-1}$  of 3.37 – Lake Vättern) and large catchments ( $A_{catchment} A_{lake}^{-1}$  of 20.94 – Lake Mälaren), reflecting the diverse hydrological characteristics of the study. Validation against observed streamflow data for these representative lakes (Figure 6B; Table S3) confirmed the ability of the scaled simulations to match not only reference data, but also observed data. Seasonal-scale performance was slightly lower (KGE of 0.46±0.21) due to timing errors, compared to stronger annual-scale performance (KGE of 0.70±0.15), indicating that the method effectively captures long-term hydrological trends."

**Comment 3:**

The validation method assumes negligible contribution of lake evaporation/precipitation compared to inflow/outflow budget. The assumption would be justified if supported by characteristic values of monthly/annual evaporation from the six lakes. Reply:

Indeed, the validation against observed data did not include the atmospheric water exchange over the lake surface (precipitation and evaporation), since we compared scaled lake inflow with observed lake outflow. We therefore estimated the potential atmospheric water exchange for the ten lakes included in this comparison. Potential evapotranspiration (PET, cm) was estimated using the empirical equation proposed by Hamon (1961), assuming that evaporation from a water surface is similar to potential evapotranspiration:  $PET = \frac{0.021 \cdot H \cdot e_s}{T_{air}}$

$$PET = \frac{0.021 \cdot H \cdot e_s}{T_{air}}$$

where H is the number of daylight hours per day,  $e_s$  is the saturated water vapor pressure (mbar) and  $T_{air}$  is daily air temperature (°C). When  $T_{air} \le 0$ , *PET* is assumed to be 0.

The saturated water vapor pressure  $(e_s)$  was calculated following Bosen (1960)

 $e_s = 33.8639 \cdot [(0.00738 \cdot T_{air} + 0.8072)^8 - 0.000019 \cdot (1.8 \cdot T_{air} + 48) + 0.001316]$ PET was calculated for the 10 lakes with available outflow observations for the period 1981-2010, using observed climate-related forcing data from the GSWP3-W5E5 climate forcing data set (Cucchi et al., 2020; Lange et al., 2021; Zhao et al., 2022) provided by ISIMIP3a. In addition, we calculated average PET, precipitation (P), the net balance P-PET and the contribution of P-PET to the lake water balance, which was then compared with streamflow inputs to assess their relative importance in lake hydrology (Table 2).

For the majority of the lakes, the atmospheric water exchange over the lake surface, expressed as P-PET, contributed less than 2% of the streamflow inputs, confirming that evaporation and precipitation can be considered negligible when comparing simulated streamflow inflows with

observed outflows. However, for lakes with long water residence time, such as lakes Vänern and Vättern, residence times of 9.8 and 58 years respectively (Kvarnäs, 2001), the *P–PET* contribution was higher, approximately 22 % and 8.5 % respectively, reducing the accuracy of the comparisons in these two particular lakes.

Table 2. *PET*, *P*, *P-PET* and % contribution to *Q*.

| Lake   |             | PET                      | P                        | P-PET                    | % contribution |
|--------|-------------|--------------------------|--------------------------|--------------------------|----------------|
|        | Name        | (mm year -1 ) | (mm year -1 ) | (mm year -1 ) | to Q           |
| 102    | Mälaren     | 595.55                   | 655.51                   | 59.96                    | 2.05           |
| 104    | Vättern     | 579.53                   | 741.82                   | 162.29                   | 22.33          |
| 105    | Vänern      | 588.39                   | 838.67                   | 250.28                   | 8.51           |
| 1150   | Siljan      | 520.26                   | 734.40                   | 214.14                   | 1.98           |
| 12423  |             | 481.19                   | 710.76                   | 229.57                   | 0.39           |
| 12791  |             | 537.47                   | 741.36                   | 203.89                   | 0.71           |
| 12809  | Erken       | 596.88                   | 628.67                   | 31.79                    | 2.03           |
| 12965  | Roxen       | 594.74                   | 662.11                   | 67.36                    | 0.24           |
| 142240 |             | 628.83                   | 630.56                   | 1.72                     | < 0.01         |
| 152977 | Hasselasjön | 510.18                   | 732.70                   | 222.52                   | 0.76           |

The Material and Methods section has been revised to reflect this point: "Although the observed data represent discharge downstream of the lakes (lake outflows), while the simulations estimate lake inflows, we assume that the atmospheric water exchange (precipitation and evaporation) over the lake surfaces in Sweden are relatively minor compared to total inflow and outflow volumes, particularly at monthly and annual timescales (Text S1)." Text S1, included in the supplementary material, details the calculation of the atmospheric water exchange over the lake surfaces as describe above.

**References:**

- Bosen, J. F.: A formula for approximation of the saturation vapor pressure over water, Monthly Weather Review, 88, 275–276, 1960.
- Cucchi, M., Weedon, G. P., Amici, A., Bellouin, N., Lange, S., Müller Schmied, H., Hersbach, H., and Buontempo, C.: WFDE5: bias-adjusted ERA5 reanalysis data for impact studies, Earth Syst. Sci. Data, 12, 2097–2120, https://doi.org/10.5194/essd-12-2097-2020, 2020.
- Hamon, W. R.: Estimating potential evapotranspiration, Journal of the Hydraulics Division, 87, 107–120, 1961.
- Kvarnäs, H.: Morphometry and Hydrology of the Four Large Lakes of Sweden, AMBIO: A Journal of the Human Environment, 30, 467–474, 2001.
- Lange, S., Menz, C., Gleixner, S., Cucchi, M., Weedon, G. P., Amici, A., Bellouin, N., Müller Schmied, Hans Hersbach, Buontempo, C., and Cagnazzo, C.: WFDE5 over land merged with ERA5 over the ocean (W5E5 v2.0), https://doi.org/10.48364/ISIMIP.342217, 2021.
- Zhao, G., Li, Y., Zhou, L., and Gao, H.: Evaporative water loss of 1.42 million global lakes, Nat. Commun., 13, 3686, https://doi.org/10.1038/s41467-022-31125-6, 2022.

---

## Author Response (AR1)

Ayala, A. I., Hinostroza, J. L., Mercado-Bettín, D., Marcé, R., Gosling, S. N., Pierson, D. C., and Sobek, S.: Integration of the Global Water and Lake Sectors within the ISIMIP framework through scaling of streamflow inputs to lakes, EGUsphere [preprint], https://doi.org/10.5194/egusphere-2025-3126, 2025.

**Reviewer 1 (Miaohua Mao)**

Summary:

This work integrates the stream flows from the nearby catchments into 71 lakes in Sweden, based on the scaling method of the global water and lake sector model. The model performances are compared with referenced model results and observed data from stations. The authors finally conclude that the updated model is satisfactory on modeling the streamflow. The authors have done a good work in explaining the workflow of their coded work, while the reviewers have some comments and suggestions needed to be clarified before it can be published after Minor Revision.

We thank the reviewer for the constructive comments and thoughtful suggestions, which have helped us improve the quality and clarity of the manuscript. Below, we provide a detailed response to each comment, and we indicate how the manuscript has been revised accordingly.

Comments:

Comment 1:

The reviewer's suggestion is avoiding using the specific values for the KGE in this Abstract section. Instead, this section should provide epitome of the entire work in a succinct and clear way.

Reply:

We have revised the Abstract by removing the specific *KGE* values rephrasing the content to provide a more general summary of main findings of the study. The part containing the *KGE* values was replaced by the following revised sentences: "The methodology was applied to 70 lakes across Sweden covering a wide range of sizes, hydrological settings and catchment characteristics. The estimated streamflow was validated against both the streamflow outputs from the hydrological model HYPE and observed data. The comparison demonstrated good agreement in terms of long-term streamflow mean and seasonal pattern, indicating that the proposed approach is capable of producing reliable streamflow estimates without requiring high-resolution local models." (Lines 25-31 track changes file)

Comment 2:

The authors have done a good work in introducing the previous study work and its research gap, and what they need to do to fill this research gap, i.e., develop the coupled streamflow and lake model via the various discharges (e.g., surface, subsurface, groundwater etc.)

Reply:

We thank the reviewer for this positive feedback. We are pleased that the research gap and motivation of our study were clear and well received.

Comment 3:

Material and methods. This section is generally well written and Fig. 2, 3, and 4 are nice figures to illustrate the procedure of the modeling frame well. Regarding the Section 2.5 Validation of streamflow at catchment scale, it is better by providing the range for the quality of Kling-Gupta efficiency (KGE) values. For example, in which ranges stand for model performance is excellent, good, poor etc., and this definition needs some references to support it. Another

comment is to define the CVsim and CVobs, which the reviewer considers as Coefficient of Variation.

Reply:

We appreciate the reviewer's positive feedback on the Material and Methods section and the figures illustrating the modeling framework.

Regarding the suggestion to provide interpretation range for the Kling-Gupta Efficiency (*KGE*), we ha now included the following classification, based on Knoben et al. (2019):

| | |
|---|---|
| *KGE=1* | Perfect |
| *0.75≤KGE<1* | Very good |
| 0.5≤KGE<0.75 | Good |
| 0.25≤KGE<0.5 | Acceptable |
| KGE<0.25 | Poor |

This classification has been added to Section 2.5 of the manuscript, along with the appropriate reference.

In addition, we clarify that the *KGE* was calculated using the *KGE()* function from the hydroGOF R package, with method="2012" to follow the revised formulation proposed by Kling et al. (2012). In this version, variability is represented by $KGE_g$, defined as the ratio of the coefficient of variation of the simulated values to the observed values:

$$KGE = \frac{CV_{sim}}{CV_{obs}} = \frac{\sigma_{sim}/\mu_{sim}}{\sigma_{obs}/\mu_{obs}}$$

where $\sigma$ and $\mu$ denote the standard deviation and mean of the simulated and observed time series respectively.

We also note that the definition of the coefficient of variation (CV) is already provided in Equation 3 of section 2.5.

The *KGE* classification has been incorporated into the Material and Methods section as follows: "Based on Knoben et al. (2019), *KGE* is interpreted as: *KGE*=1 perfect agreement, 0.75≤*KGE*<1 very good performance, 0.50≤*KGE*<0.75 good performance, 0.25≤*KGE*<0.50 acceptable performance and *KGE*<0.25 poor performance." (Lines 246-248 track changes file)

Comment 4:

Line 229: '…… we assume than lake evaporation ……' maybe changed to '…… we assume that lake evaporation ……'

Reply:

Thank you for noticing this typographical error. It has been corrected as suggested in the revised manuscript. (Line 234 track changes file)

Comment 5:

Line 223 and other places: The authors please make sure that whether 70 or 71 lakes in Sweden are studied. This needs to be consistent throughout the texts.

Reply:

A total of 71 lakes in Sweden were initially considered. However, streamflow simulations for lake 149288 were included in the validation against observations, but not in the validation against reference values due to data limitations. This discrepancy caused some confusion in the lake count in the different sections of the text.

We have now removed lake 149288 from the analysis entirely, as it was only partially included in the original validation. Consequently, the total number of lakes studied is now 70. We have carefully reviewed and revised the manuscript to ensure that this number is consistent

throughout the text and have removed all reference to lake 149288 (Lines 24, 99, 147, 229-230, 315, 352, 479 track changes file).
In addition, Figure 1, Figure 6, Figure S2, Table S1, Table S3 have been updated to exclude lake 149288. The Study sites section (Material and Methods) was revised to updated lake characteristics accordingly. In the Result section, the performance of simulated and observed streamflow for the lake 149288 was removed (lines 324-333 track changes file). The Discussion section was also revised to exclude lake 149288 (Lines 388-410, 451-454, track changes file). The Supplementary material figure comparing simulated, reference and observed streamflow for the lake 149288 was removed.

Comment 6:
Line 248-249: 'For all study sites, the KGE exceeded -0.41, indicating that the simulated streamflow provided added value compared to using long-term mean values.' The reviewer is a little bit confused that a negative value of KGE (e.g., -0.41) means this revision provides added value.
Reply:
Negative *KGE* values generally indicate poor model performance, the original intent of the sentence was to highlight that the model still performed better than a simple baseline (e.g. using the mean observed discharge as prediction).
To avoid confusion, we have revised the sentence for clarity: "For all study sites, the *KGE* exceeded -0.41, indicating that the simulated streamflow provided added value compared to simple prediction based on the long-term mean streamflow." (Line 257 track changes file).

Comment 7:
In the Results Section. The authors have compared their model with reference results and observed data, respectively. The reviewer considers that would that also necessary to compare the reference results with the observed data. By doing so, the reader could have better understood that whether the developed model improves its performance or not, compared with the traditional model (i.e., the reference results).
Reply:
Lake 149288, which had streamflow observations available but lacked reference data, was excluded from the original validation analysis to maintain consistency throughout the text. To ensure comprehensive comparison, additional lakes with both streamflow observations and reference data were included in the analysis. As a result, streamflow simulations were compared with observations for a total of 10 lakes.

The Validation of streamflow at catchment scale section (Material methods) was updated to include the new lake (lake 12423, lake 12791, Roxen – 12965, lake 142240, Hasselasjön – 152977) (Lines 227-230 track changes file). In addition, Figure 6, Figure S2, Table S3 have been updated to include these lakes. In the Result section, the performance of simulated and observed streamflow for these lakes has been added (lines 324-333 track changes file). The Discussion section was also revised to reflect the inclusion of these lakes (Lines 388-422, 451-454, track changes file). Furthermore, the Supplementary material has been updated to include figures comparing simulated, reference and observed streamflow for these lakes were included.

The Code availability and Data availability sections have also been updated to include these five additional lakes.

Additionally, in the revised results sections, we have incorporated a comparison between the reference and observed streamflow, using the same performance metrics. Note that, while

observations were available for 10 lakes, comparison were conducted for only 9 lakes, as reference streamflow data are available from 1981-2010. A new table (Table S4) presenting performance metrics has been added to the supplementary material and a brief description of the performance has been added to the result section.

Brief description of the performance added to the result section: "Finally, a further evaluation was conducted by comparing reference and observed streamflow for 9 study sites (note that the reference and observations datasets cover different time periods, which limited direct comparability in the 10 study sites for which observations were available) (Table S4). At the monthly scale, the average $KGE$ was 0.44±0.44 (with $KGE_r$ of 0.65±0.23, $KGE_b$ of 1.12±0.34, $KGE_g$ of 1.13±0.46), indicating on average acceptable agreement with substantial inter-site differences. At the yearly scale, performance improved to $KGE$ of 0.55±0.26 (with $KGE_r$ of 0.78±0.12, $KGE_b$ of 1.12±0.34, $KGE_g$ of 0.77±0.19). Overall, these results demonstrate that the scaling method provides added value, improving the simulations of streamflow compared with standard catchment-scale hydrological models." (Lines 334-340 track changes file)

Comment 8:
The current writing of this part could be improved in a more detailed way. For example, providing some skill metric values that are specified (e.g., various KGE values), so that this work could be better summarized in a more strict way.
Reply:
We have revised the Results section to improve the clarity and structure of the performance evaluation. These changes make the summary more quantitative and structured, as suggested, and improve the overall readability of the results. The changes can be found throughout the revised results section. Furthermore, as show in the Supplementary Material (Tables S2-S4) additional performance metrics including $MBE$, $RMSE$, $NRMSE$ and $NSE$ are already provided for each study site. We believe these revisions address your concerns and improve the overall presentation of the results.

Revised result section (Lines 252-342 track changes file):

[revised manuscript text omitted]

**Reviewer 2**

Summary:
The study presents a method of re-scaling gridded water flow data on lake catchments. The method is developed within the framework of ISIMIP, the model intercomparison platform facilitating access to climate scenarios, models, and observational data for model validation. The manuscript is well-structured, clearly written, and addresses an important gap in coupling water flow and lake models on global scales. The proposed method uses a straightforward rescaling algorithm, differentiating between three options---the catchment is smaller than a single grid cell, the catchment is larger than, but the lake is smaller than a grid cell, and the lake is larger than a single grid cell. The approach has been validated against the long-term outputs of the operational regional hydrological model HYPE applied to 71 Swedish lakes and against a smaller observational dataset, demonstrating satisfactory performance. The results, summarized in two pages and two figures, are clear and concise. The impact on the modeling community can be however limited: while Swedish lakes provide a robust and diverse test case, the extrapolation to global conditions (particularly arid and tropical systems with highly variable evaporation and different hydrological regimes) remains speculative. Still, it is a valuable methodological contribution, with openly available code and datasets, which ensures reproducibility, and an initial step towards coupling lake and water flow modeling in climate models.

We thank the reviewer for the positive and constructive feedback. Regarding the concern about extrapolation to global conditions, we would like to clarify that our study focuses on rescaling streamflow inputs into lakes. All hydrological calculations are taken from the global hydrological model WaterGAP 2, which has been extensively validated a cross a range of climatic and hydrological regimes, including arid and tropical systems and thus encompassing variable hydrological and evaporation regimes. Our method does not perform new hydrological modeling but operates on the existing generated by WaterGAP 2, with the purpose to scale them to lake catchments. Therefore, the applicability of our approach globally relies on the underlying WaterGAP 2 outputs, not on the rescaling approach itself. The validation of our scaling approach was conducted on a wide variety of lake and catchment properties, particularly in terms of size, suggesting its suitability for global application.

The following sentences were removed from the Discussion section to avoid confusion: "While the Swedish climate is temperate to subarctic, factors such as evaporation may differ in arid and tropical conditions. Thus, although climate-related refinements may be necessary for certain regions, the core method grounded in topographic and geometric scaling is broadly applicable." (Lines 375-377 track changes file)

Comments:
Comment 1:
The case of Lake Mälaren demonstrates that irregular morphologies can strongly affect scaling performance. The authors might consider providing more concrete recommendations for how to approach such cases practically.
Reply:
The case of Lake Mälaren indeed highlights the impact of irregular morphologies on scaling performance. However, despite the lake's very complex shape and bathymetry, the modeling results were still satisfactory. Specifically, for Lake Mälaren, Approach I.b yielded a good performance with a *KGE* of 0.71, while Approach II showed acceptable performance with a *KGE* of 0.47. These results demonstrate that even in lakes with complex morphologies, both approaches can deliver at least acceptable performance.
Moreover, when comparing these results to other lakes (Manuscript: Figure 6 and Table S2), Lake Mälaren is not an outlier. Several other lakes with less complex shapes showed similar performance metrics, indicating that while morphology can influence predictive performance, it is not the sole determinant of success. This suggests that practical application of the scaling approaches remains viable even in morphologically complex systems.

The Discussion section has been revised to reflect these points: "In contrast, for Lake Mälaren, which has a highly irregular shape (Figure S1), the choice of scaling approach significantly affected performance. The better performance of Approach I.b (*KGE*=0.71) compared to Approach II (*KGE*=0.47) highlights the importance of accounting for complex lake morphologies in streamflow scaling. Nevertheless, both scaling approaches achieved satisfactory performance comparable to other lakes with less complex morphologies, indicates that, although lake morphology can influence performance, it is not the sole determining factor, further supporting the robustness and practical applicability of the scaling approaches even for lakes with complex morphologies." (Lines 437-442 track changes file)

Comment 2:
Only six lakes are compared against observed streamflow. While this is understandable due to data availability, a short description of the lakes representativity, in terms of lake size, geographical location, hydrological regime, would strengthen confidence.
Reply:
The observed streamflow records were extended to 10 lakes, which represent a diverse range of physical and hydrological characteristics. Geographically, these lakes are distributed across latitudes from 58.33° to 66.66°, covering southern, central and northern regions of Sweden (Table 1). The lake area spans three orders of magnitude from 7.68 km$^2$ (lake 142240) to 5486.23 km$^2$ (lake Vänern), with catchment areas that vary independently of lake size ($A_{catchment}$ raged from 138.70 km$^2$ to 48421 km$^2$). This includes both small lakes with small catchments ($A_{catchment}$ $A_{lake}$$^{-1}$ of 5.99 – lake Erken) and large catchments ($A_{catchment}$ $A_{lake}$$^{-1}$ of 139.91– lake Roxen), as well as large lakes with small catchments ($A_{catchment}$ $A_{lake}$$^{-1}$ of 3.37 – Lake Vättern) and large catchments ($A_{catchment}$ $A_{lake}$$^{-1}$ of 20.94 – Lake Mälaren), reflecting the diverse hydrological characteristics of the study. Overall, despite the limited availability of observed

streamflow data, these ten lakes provide a representative cross-section of the variability in lake size, catchment characteristics and geographical distribution within the study area.

Table 1: Characteristics of the study sites with available streamflow observations.

| Lake | Name | Longitude | Latitude | $A_{lake}$ [km$^2$] | $A_{catchment}$ [km$^2$] | $A_{catchment}$ $A_{lake}^{-1}$ |
|------|------|-----------|----------|---------|-------------|------------|
| 102 | Mälaren | 16.79 | 59.49 | 1083.13 | 22682.20 | 20.94 |
| 104 | Vättern | 14.49 | 58.33 | 1888.04 | 6369.10 | 3.37 |
| 105 | Vänern | 13.55 | 58.88 | 5486.23 | 48421.00 | 8.83 |
| 1150 | Siljan | 14.77 | 60.86 | 290.88 | 12084.50 | 41.54 |
| 12423 | | 14.15 | 62.05 | 63.59 | 8357.00 | 131.42 |
| 12791 | | 15.57 | 60.07 | 34.77 | 2213.30 | 63.66 |
| 12809 | Erken | 18.60 | 59.84 | 23.14 | 138.70 | 5.99 |
| 12965 | Roxen | 15.63 | 58.49 | 94.55 | 13228.50 | 139.91 |
| 142240 | | 22.22 | 66.66 | 7.68 | 1272.30 | 165.66 |
| 152977 | Hasselasjön | 16.78 | 62.08 | 8.36 | 610.00 | 72.97 |

The Discussion section has been revised to reflect these points: "Although validation against observed streamflow is constrained due to data availability, the 10 lakes used for validation are broadly representative of the 70 lakes included in the study. Geographically, these lakes are distributed across latitudes from 58.33° to 66.66°, covering southern, central and northern regions of Sweden (Table S3). The lake area spans three orders of magnitude from 7.68 km$^2$ (lake 142240) to 5486 km$^2$ (lake Vänern), with catchment areas that vary independently of lake size ($A_{catchment}$ raged from 138.7 km$^2$ to 48421 km$^2$). This includes both small lakes with small catchments ($A_{catchment}$ $A_{lake}^{-1}$ of 5.99 – lake Erken) and large catchments ($A_{catchment}$ $A_{lake}^{-1}$ of 139.91– lake Roxen), as well as large lakes with small catchments ($A_{catchment}$ $A_{lake}^{-1}$ of 3.37 – Lake Vättern) and large catchments ($A_{catchment}$ $A_{lake}^{-1}$ of 20.94 – Lake Mälaren), reflecting the diverse hydrological characteristics of the study. Validation against observed streamflow data for these representative lakes (Figure 6B; Table S3) confirmed the ability of the scaled simulations to match not only reference data, but also observed data. Seasonal-scale performance was slightly lower (*KGE* of 0.46±0.21) due to timing errors, compared to stronger annual-scale performance (*KGE* of 0.70±0.15), indicating that the method effectively captures long-term hydrological trends." (Lines 443-454 track changes file)

Comment 3:
The validation method assumes negligible contribution of lake evaporation/precipitation compared to inflow/outflow budget. The assumption would be justified if supported by characteristic values of monthly/annual evaporation from the six lakes.
Reply:
The validation against observed data did not include the atmospheric water exchange over the lake surface (precipitation and evaporation), since we compared scaled lake inflow with observed lake outflow. We therefore estimated the potential atmospheric water exchange for the ten lakes included in this comparison. Potential evapotranspiration (*PET*, cm) was estimated using the empirical equation proposed by Hamon (1961), assuming that evaporation from a water surface is similar to potential evapotranspiration:

$$PET = \frac{0.021 \cdot H \cdot e_s}{T_{air}}$$

where $H$ is the number of daylight hours per day, $e_s$ is the saturated water vapor pressure (mbar) and $T_{air}$ is daily air temperature (°C). When $T_{air} \leq 0$, $PET$ is assumed to be 0.

The saturated water vapor pressure ($e_s$) was calculated following Bosen (1960)

$$e_s = 33.8639 \cdot [(0.00738 \cdot T_{air} + 0.8072)^8 - 0.000019 \cdot (1.8 \cdot T_{air} + 48) + 0.001316]$$

$PET$ was calculated for the 10 lakes with available outflow observations for the period 1981-2010, using observed climate-related forcing data from the GSWP3-W5E5 climate forcing data set (Cucchi et al., 2020; Lange et al., 2021; Zhao et al., 2022) provided by ISIMIP3a. In addition, we calculated average $PET$, precipitation (P), the net balance $P–PET$ and the contribution of $P–PET$ to the lake water balance, which was then compared with streamflow inputs to assess their relative importance in lake hydrology (Table 2).

For the majority of the lakes, the atmospheric water exchange over the lake surface, expressed as $P–PET$, contributed less than 2% of the streamflow inputs, confirming that evaporation and precipitation can be considered negligible when comparing simulated streamflow inflows with observed outflows. However, for lakes with long water residence time, such as lakes Vänern and Vättern, residence times of 9.8 and 58 years respectively (Kvarnäs, 2001), the $P–PET$ contribution was higher, approximately 22 % and 8.5 % respectively, reducing the accuracy of the comparisons in these two particular lakes.

Table 2. *PET, P, P-PET* and % contribution to *Q*.

| Lake | Name | PET (mm year⁻¹) | P (mm year⁻¹) | P – PET (mm year⁻¹) | % contribution to Q |
|------|------|------|------|------|------|
| 102 | Mälaren | 595.55 | 655.51 | 59.96 | 2.05 |
| 104 | Vättern | 579.53 | 741.82 | 162.29 | 22.33 |
| 105 | Vänern | 588.39 | 838.67 | 250.28 | 8.51 |
| 1150 | Siljan | 520.26 | 734.40 | 214.14 | 1.98 |
| 12423 | | 481.19 | 710.76 | 229.57 | 0.39 |
| 12791 | | 537.47 | 741.36 | 203.89 | 0.71 |
| 12809 | Erken | 596.88 | 628.67 | 31.79 | 2.03 |
| 12965 | Roxen | 594.74 | 662.11 | 67.36 | 0.24 |
| 142240 | | 628.83 | 630.56 | 1.72 | <0.01 |
| 152977 | Hasselasjön | 510.18 | 732.70 | 222.52 | 0.76 |

The Material and Methods section has been revised to reflect this point: "Although the observed data represent discharge downstream of the lakes (lake outflows), while the simulations estimate lake inflows, we assume that the atmospheric water exchange (precipitation and evaporation) over the lake surfaces in Sweden are relatively minor compared to total inflow and outflow volumes, particularly at monthly and annual timescales (Text S1)." (Lines 437-442 track changes file)

Text S1 in the Supplementary material includes the calculation of the atmospheric water exchange over the lake surfaces as describe above.